# Comparative Study on Hot Metal Flow Behaviour of Virgin and Rejuvenated Heat Treatment Creep Exhausted P91 Steel

Shem Maube [1,*], Japheth Obiko [2,*], Josias Van der Merwe [3,4], Fredrick Mwema [1], Desmond Klenam [3] and Michael Bodunrin [3,4]

[1]  Department of Mechanical Engineering, Dedan Kimathi University of Technology, Private Bag 10143, Dedan Kimathi, Nyeri 10143, Kenya
[2]  Department of Mining, Materials and Petroleum Engineering, Jomo Kenyatta University of Agriculture and Technology, Nairobi P.O. Box 6200-00200, Kenya
[3]  School of Chemical and Metallurgical Engineering, Faculty of Engineering and the Built Environment, University of the Witwatersrand, Johannesburg, Private Bag 3, Johannesburg 2050, South Africa
[4]  DSI-NRF Centre of Excellence in Strong Materials, University of the Witwatersrand, Private Bag 3, Johannesburg 2050, South Africa
*  Correspondence: smaube@gmail.com (S.M.); japheth.obiko97@gmail.com (J.O.)

**Abstract:** This article reports on the comparative study of the hot deformation behaviour of virgin (steel A) and rejuvenated heat treatment creep-exhausted (steel B) P91 steels. Hot uniaxial compression tests were conducted on the two steels at a deformation temperature range of 900–1050 °C and a strain rate range of 0.01–10 s$^{-1}$ to a total strain of 0.6 using Gleeble® 3500 equipment. The results showed that the flow stress largely depends on the deformation conditions. The flow stress for the two steels increased with an increase in strain rate at a given deformation temperature and vice versa. The flow stress–strain curves exhibited dynamic recovery as the softening mechanism. The material constants determined using Arrhenius constitutive equations were: the stress exponent, which was 5.76 for steel A and 6.67 for steel B; and the apparent activation energy, which was: 473.1 kJ mol$^{-1}$ for steel A and 564.5 kJmol$^{-1}$ for steel B. From these results, steel A exhibited better workability than steel B. Statistical parameters analyses showed that the flow stress for the two steels had a good correlation between the experimental and predicted data. Pearson's correlation coefficient (R) was 0.97 for steel A and 0.98 for steel B. The average absolute relative error (AARE) values were 7.62% for steel A and 6.54% for steel B. This study shows that the Arrhenius equations can effectively describe the flow stress behaviour of P91 steel, and this method is applicable for industrial metalworking process.

**Keywords:** P91 steel; rejuvenation heat treatment; creep exhausted; constitutive equation; hot deformation

## 1. Introduction

P91 steel belongs to a family of 9–12% Cr ferritic creep-resisting steels [1]. This steel was initially developed in the 1970s for application in steam generators of fast breeder reactors by Oak Ridge National Laboratory [2,3]. At present, P91 steel has wide-ranging applications in high-temperature and high-pressure components such as headers, heat exchangers, water-wall tubing, and piping systems of fossil-fuel power plants [4,5]. The suitability of this steel for these applications is due to its superior thermo-mechanical properties that include high corrosion resistance, good weldability, and low thermal expansion. It also exhibits excellent fatigue strength, toughness and good creep performance [6]. P91 steel is commonly used in the temperature range of 540 °C to 650 °C. Some of its mechanical properties conforming to ASTM A335 are a minimum tensile strength of 585 MPa, a minimum yield strength of 415 MPa, and elongation of 19.5%. At room temperature, it has a hardness of HB 250 and a minimum creep rupture strength of 69 MPa at 650 °C [7].

After prolonged service, steel experiences internal damage due to microstructural changes from long-term exposure to high temperatures under stress. These changes are

detrimental to mechanical properties, especially toughness, and cause degradation of the creep properties [8]. This leads to the replacement of degraded components at the end of the design lifetime. The high cost of replacement has increased research interest in alternative methods of extending the components' service life. The implementation of an appropriate rejuvenation processes has the potential to significantly extend the operational service time of degraded components.

The rejuvenation of degraded metals and alloys can be achieved through various thermal procedures such as heat treatment and hot isostatic pressing [9]. These processes can modify the material's microstructure, thus improving the mechanical properties of damaged or exhaust components [10]. Proper heat treatment schedules cause grain refinement and can re-establish the original microstructure and thus the material properties of service-exhausted components [11]. Information in the literature about the effect of rejuvenation heat treatment on metal flow behaviour, especially P91 steel, is scarce, and therefore there is a need for further research.

In 9–12% Cr ferrite steels, heat treatment is generally performed by means of normalising, cooling and tempering processes [12]. Normalising is conducted above the upper critical temperature, $Ac_3$, and tempering is conducted below the lower critical temperature, $Ac_1$ [13]. Various authors [2,13–15] have applied different normalisation and tempering temperatures for P91 steel within the $Ac_3$ and $Ac_1$ requirements. Pandey et al. [2] reported the phase transformation temperatures of P91 as 810 °C to 825 °C ($Ac_1$) and 912 °C to 930 °C ($Ac_3$). According to Abe [16], the $Ac_1$ was 800 to 830 °C and $Ac_3$ 890 to 940 °C in P91. The variations in heat treatment temperatures of P91 are due to minor differences in chemical composition (especially Ni + Mn), rate of heating and prior austenite grain boundaries (PAGBs) [14]. The American Society of Mechanical Engineers (ASME) [16] recommended an austenitisation temperature of 1040 to 1080 °C and tempering at 730 to 780 °C in P91 pipes.

In 9–12% Cr ferritic steels, creep deformation mechanisms are mainly associated with changes in the microstructure. These microstructure changes cause a reduction in dislocation density, the migration of sub-grain boundaries, the formation of new secondary phases (Laves and Z-phase) and the growth of sub-grain structure [17,18]. During creep, the sub-grains grow, causing a decrease in the creep strength of the 9–12% Cr steels. The lath phase coarsens around $M_{23}C_6$ (M = chromium, iron, tungsten and molybdenum) carbides along the PAGBs, reducing their pinning effect and hence reducing the strengthening mechanism [19].

Heat treatment of P91 steel produces a tempered lath martensite structure with $M_{23}C_6$ and MX (M = Niobium, Vanadium and X-carbon) precipitates [2]. During normalisation, the homogenisation of the microstructure occurs, causing an untempered lath microstructure with a small number of precipitates and a high dislocation density. Tempering causes the precipitation of the $M_{23}C_6$ carbides along the grain boundaries and MX particles in the matrix [19]. The high dislocation density and the $M_{23}C_6$ precipitates are responsible for the pinning of lath boundaries, while MX precipitates prevent dislocation movement [2,17]. The heat treatment process, therefore, restores the effects of the creep process and provides a stable microstructure for the steel to achieve high creep strength [20].

Rejuvenation heat treatment, as mentioned earlier, provides an economic benefit in terms of the expensive process of replacing parts and time and energy savings [9]. In addition, an appropriately designed refurbishment process can significantly improve the lifetime service of components or a production plant. To sufficiently develop a heat treatment process, it is imperative to understand the response of heat treatment on creep-exhausted steel to thermomechanical processing parameters. This study provides a basis for understanding the metal flow pattern during hot deformation. A comparative study using constitutive models developed from the hot deformation process of a virgin (Steel A) and creep-exhausted steel (Steel B) is used for this analysis [21].

Phenomenological constitutive models such as Arrhenius equations are widely used to study the metal flow behaviour of various metals and alloys [22]. Some examples of such

materials studied using this model include low-carbon steel [23,24], P92 steel [25], nickel-based super alloys [26], alloys of aluminium [23], and magnesium [27]. The accuracy and reliability of this model in predicting the flow stress forms the basis of studying the material process behaviour [28]. These models also act as inputs for finite element method codes. These computer codes assist in the simulation and the optimisation of thermomechanical process parameters. The simulation results depend on the accuracy and dependability of the constitutive models [21,28].

The objective of this present study was to investigate the flow stress behaviour of steel A and steel B using an Arrhenius constitutive model. The steel B was heat treated by normalisation followed by air cooling, then tempering processes. Isothermal hot deformation tests were then conducted at various strain rates and temperatures to study the flow stress behaviour for the two steels. Experimental flow stress values were used to develop constitutive equations based on the Arrhenius-type equation. These equations were validated using statistical parameters.

## 2. Materials and Methods

The chemical composition of the P91 steels studied is given in Table 1. The samples for hot compression were machined from the two steels into a cylindrical shape with a diameter of 8 mm and a height of 12 mm. The heat treatment process for P91 steel according to various studies [3,19,29] was recommended as: normalising at a temperature between 1030 °C and 1080 °C for 20 to 40 min, then air cooling followed by tempering between 730 °C and 780 °C for 1 to 2 h. In this study, the steel B samples were normalised at 1050 °C for 40 min, air cooled and then tempered at 760 °C for 2 h, as illustrated in Figure 1. Figure 2a,b show the optical micrographs of undeformed steel A and steel B. The micrograph of steel A shows tempered martensitic microstructure with prior austenite grain boundaries (PAGBs), martensite lath boundaries and the dispersion of fine precipitates in the matrix and over boundaries. This is a typical microstructure reported in various studies of P91 steel [2,30,31]. The microstructure of the steel B was modified as a result of service conditions, with the grain boundaries being more apparent and an equiaxed grain structure [9,32].

**Table 1.** Chemical elements (wt %) of the P91 steels.

| Cr | C | Mn | Mo | V | Nb | W | Ni | Si | Fe | P | Cu | Mg | Sn |
|---|---|---|---|---|---|---|---|---|---|---|---|---|---|
| 9.189 | 0.1 | 0.447 | 0.885 | 0.191 | 0.076 | <0.01 | 0.158 | 0.254 | 88.01 | 0.02 | 0.086 | 0.016 | 0.006 |

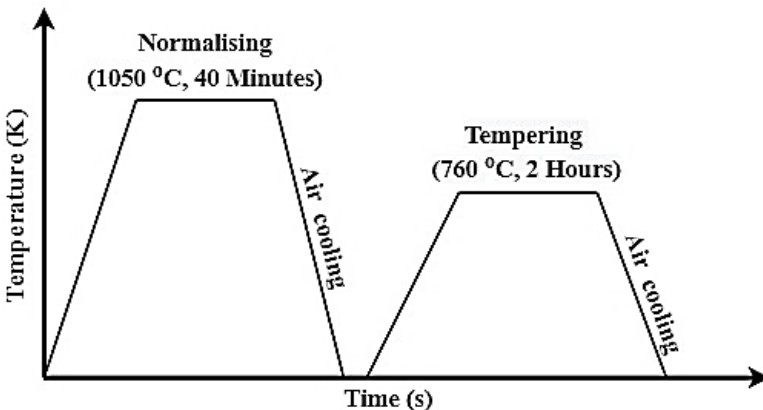

**Figure 1.** The heat treatment process of creep exhausted P91 steel.

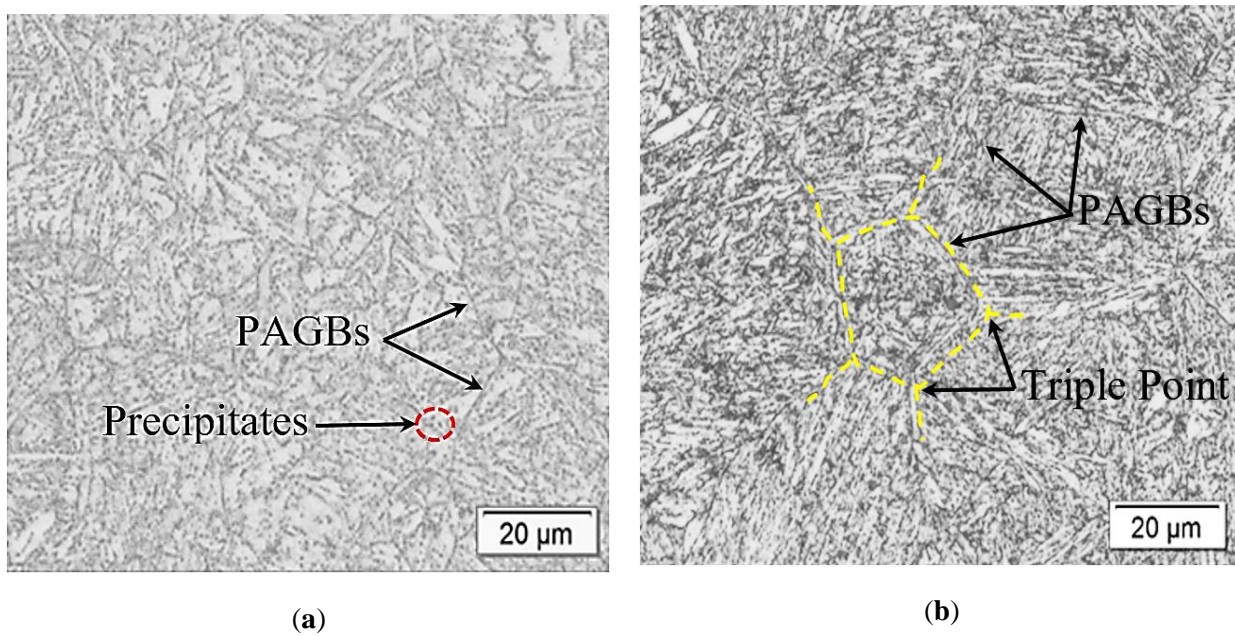

(**a**)                                                    (**b**)

**Figure 2.** The optical micrographs for (**a**) steel A and (**b**) steel B.

The uniaxial hot compression parameters using Gleeble® 3500 equipment were as follows: deformation temperatures of 900 °C, 950 °C, 1000 °C, and 1050 °C and strain rates of $0.01\ s^{-1}, 0.1\ s^{-1}, 1\ s^{-1}$, and $10\ s^{-1}$. The samples were deformed to a total strain of 0.6. Before testing, an R-type thermocouple was welded at the midpoint on the samples to monitor temperature during the deformation process. Figure 3 is a schematic diagram illustrating the thermal deformation process used in this study. All the specimens were heated at 5 °C/s to 1100 °C and held isothermally for 180 s before cooling to the deformation temperature.

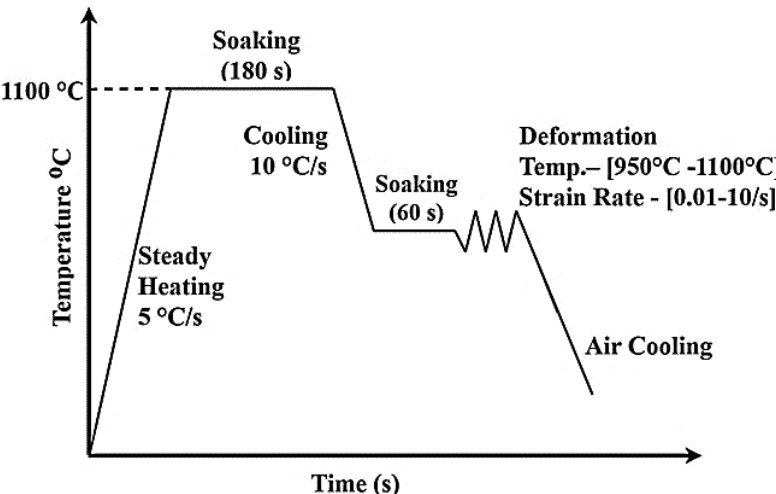

**Figure 3.** Thermal deformation profile.

The deformed samples were cut along the compression axis for microstructural analysis. The specimen was prepared following the metallographic procedures. Etching was performed using Villella's reagent solution (1 g Picric acid + 5 mL HCl + 100 mL ethanol).

## 3. Results

### 3.1. Flow Stress–Strain Curves

Flow stress–strain curves provide information on metal deformation behaviour during forging. However, the accuracy of the curves is affected by the interfacial friction between the tool and the workpiece as it modifies the force required during deformation [33]. At the sample and the die end, graphite foil and nickel paste were placed to reduce the effect of friction during deformation. At high strain, however, the friction effect is more significant, causing inhomogeneous deformation and hence contact between the samples and die surface. This is known as barrelling, and the flow stress–strain curves have to be corrected for friction before further analysis [33,34]. The barrelling coefficient was calculated using the sample geometry size according to Roebuck et al. [35]. Equation (1) was used to determine the barrelling coefficient, $b$, with the allowable range of barrelling coefficients as $1 < b \leq 1.1$ [36]. Friction within this range is negligible and only stress–strain curves of samples with a barrelling coefficient beyond this range were corrected for friction [37]. Figure 4 shows plots of uncorrected and friction-corrected flow stress curves of the two P91 steels at a temperature of 950 °C and strain rate of 0.01 s$^{-1}$, 0.1 s$^{-1}$, 1 s$^{-1}$, and 10 s$^{-1}$. The results show that the measured flow stress values in the stress–strain curves were higher than the friction-corrected values. This variation may be due to the interfacial friction effect experienced at the test sample and anvil interface.

$$b = \frac{h_f d_f^2}{h_o d_o^2} \tag{1}$$

where $h_o$ and $h_f$ are the initial and final height of the sample and $d_o$ and $d_f$ are the initial and final diameter of the sample.

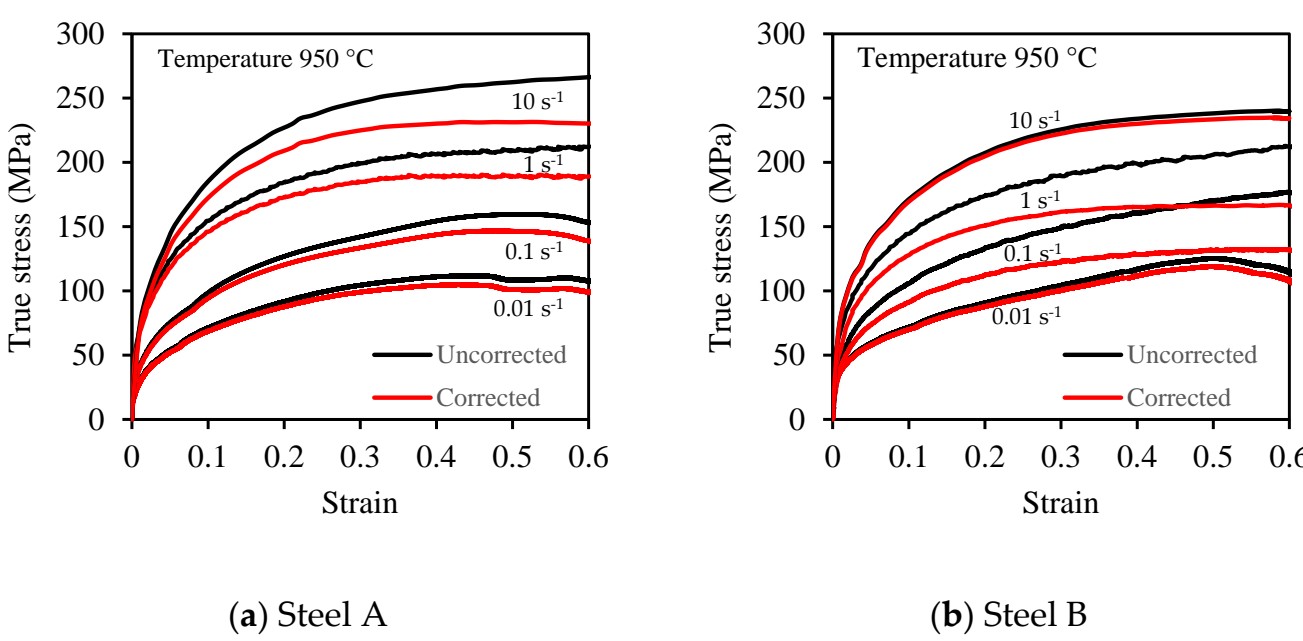

**(a)** Steel A        **(b)** Steel B

**Figure 4.** Plots of uncorrected and corrected flow stress–strain curves.

Figure 5 (steel A) and Figure 6 (steel B) show friction-corrected flow stress–strain curves at various deformation conditions. The flow stress–strain curves show that the flow stress depends on the strain rate and deformation temperature. For a given strain rate, flow stresses decrease with an increase in the deformation temperature of the two steels investigated. These metal flow patterns may be due to the increase in the mobility of grain boundaries as deformation temperature increases [38]. At a given temperature, flow stress increases with an increase in the strain rate. The flow stress–strain curves

(Figures 5 and 6) show that at the beginning of the deformation process, the flow stress increases rapidly up to a strain of ~0.2. This characteristic flow behaviour occurs due to the rapid generation of dislocation density resulting in a work-hardening deformation mechanism [39]. At higher strain (>0.2), most flow stress–strain curves for the two steels reached a steady-state condition. This characteristic behaviour of flow curves indicates a balance between work-hardening and -softening mechanisms, especially dynamic recovery (DRV) for these steels [40]. The flow stress–strain curves for the two steels studied did not exhibit a clear peak and subsequent steady-state flow stress caused by DRX [36,41]. Therefore, for most deformation conditions, the flow curves had increasing flow stress until the start of the saturation stress ($\sigma_{sat}$) region. The results show that DRV involved the softening mechanism. Similar flow stress–strain curve behaviour results are available in the literature for creep-resistant steels [42,43].

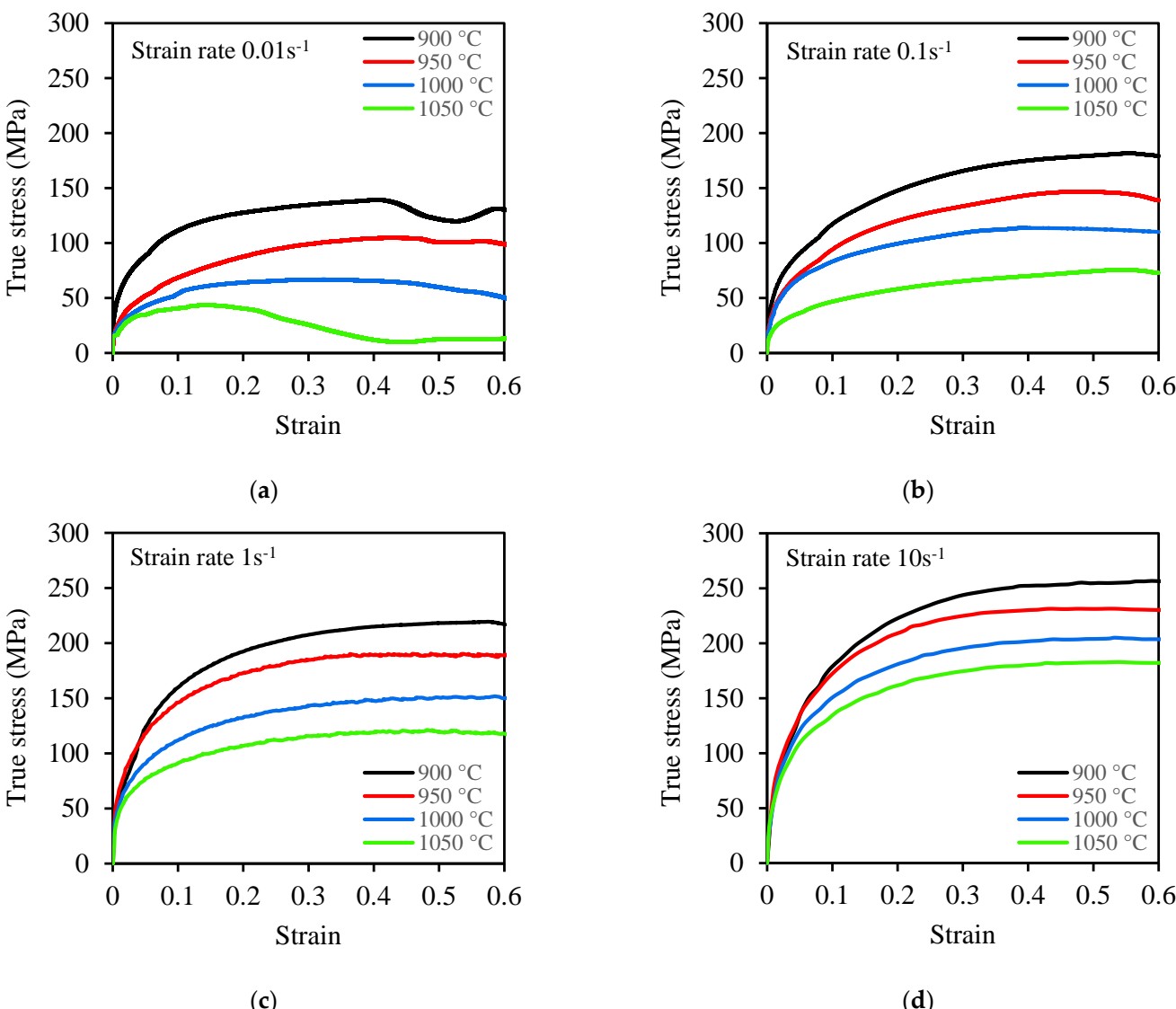

**Figure 5.** Flow stress–strain curve of steel A.

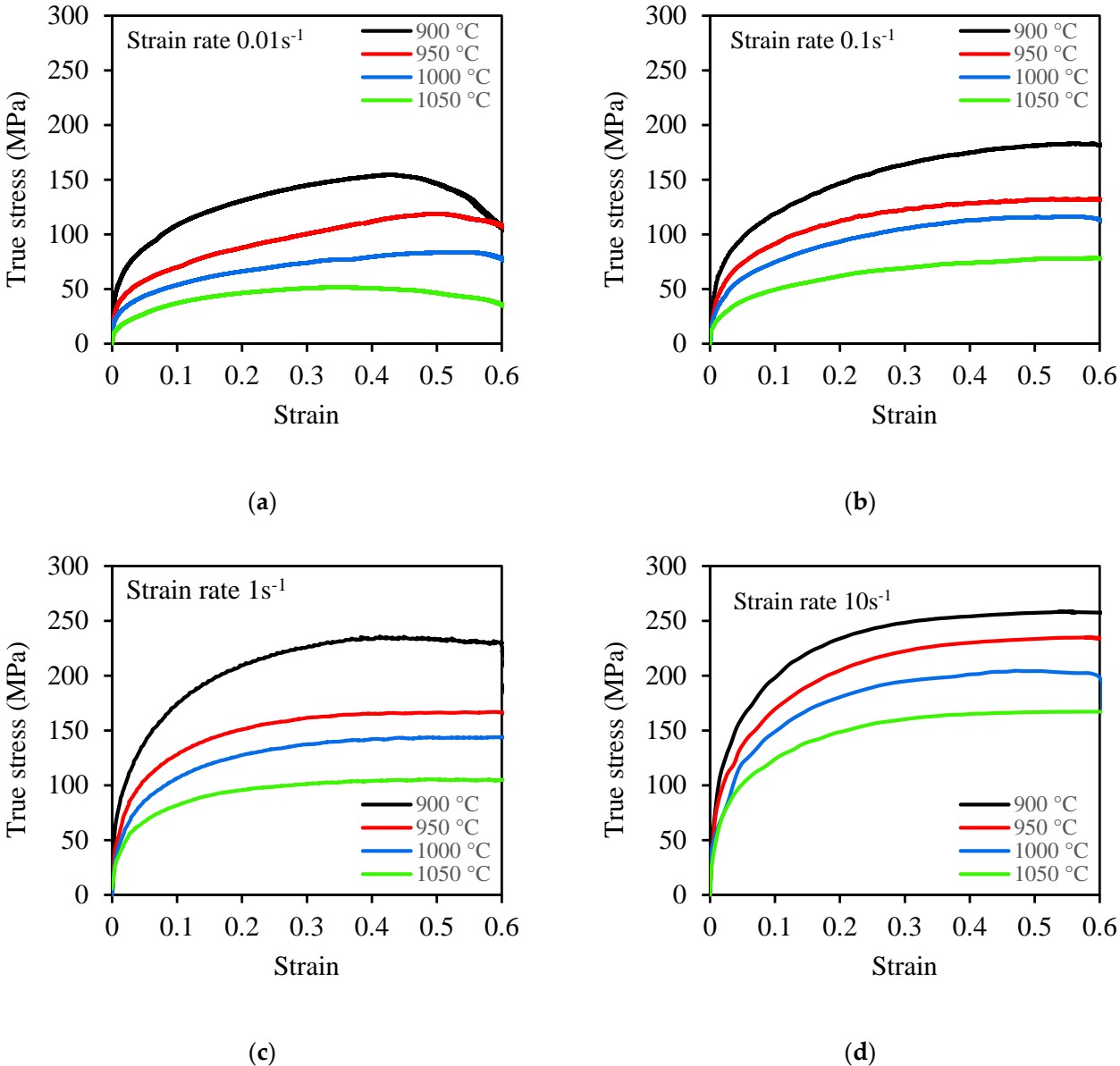

**Figure 6.** Flow stress–strain curves of steel B.

*3.2. Constitutive Equation and Material Constants*

Constitutive equations relate the flow stress and hot deformation parameters (strain rate, strain, deformation temperature) during the hot working process. Various equations have been proposed to predict the flow stress behaviour of alloys and metals during hot deformation. Some examples include the Johnson–Cook equation, the Zerilli–Armstrong equation, the Hensel–Spittel equation and the Arrhenius-type equation. These equations have varying accuracy in predicting the flow stress and deformation conditions [44,45]. The Arrhenius type equation (Equation (2)) has wide-ranging applications in analysing the metal flow behaviour during deformation. Therefore, in this study, this model was used to compare the flow stress behaviour of the two steels [22,41,46,47]. The Zener–Hollomon parameter (Z) in Equation (2) was also used to represent the effect of temperature and strain rate [48].

$$\dot{\varepsilon} = Af(\sigma)\exp\left(\frac{-Q}{RT}\right) \tag{2}$$

$$Z = \dot{\varepsilon}\exp\left(\frac{Q}{RT}\right) \tag{3}$$

where $\sigma$ is the flow stress, $\dot{\varepsilon}$ is the strain rate, $T$ is the absolute temperature (K), $Q$ is the activation energy for hot deformation (kJ/mol), $A$ is a material constant and $R$ is the universal gas constant ($8.314 \ \mathrm{J mol^{-1} \ K^{-1}}$).

$$Z = \dot{\varepsilon}\exp\left(\frac{Q}{RT}\right) = f(\sigma) = A\sigma^{n'} \tag{4}$$

$$Z = \dot{\varepsilon}\exp\left(\frac{Q}{RT}\right) = f(\sigma) = A\exp(\beta\sigma) \tag{5}$$

$$Z = \dot{\varepsilon}\exp\left(\frac{Q}{RT}\right) = f(\sigma) = A[sinh(\alpha\sigma)]^{n} \tag{6}$$

The relationship between the Zener–Hollomon parameter and flow stress is expressed by Equations (4) to (6) [49]. The terms in these equations—$n'$, $\beta$, $n$, and $\alpha$—are material constants. Equation (4) is the power law applied for low stresses ($\alpha\sigma < 0.8$), as it breaks down at higher flow stress. Equation (5) is the exponential law used for higher stress ($\alpha\sigma > 1.2$) and breaks down at a strain rate under $1 \ \mathrm{s^{-1}}$ and high temperature [50]. Equation (6) is a hyperbolic sine function equation developed by Sellars and Tegart [51] to be applicable to a wide range of stresses (both high and low). Equations (4)–(6) can also give Equations (7) and (9) [25]:

$$\dot{\varepsilon} = A\sigma^{n'}\exp\left(\frac{-Q}{RT}\right) \tag{7}$$

$$\dot{\varepsilon} = A\exp(\beta\sigma)\left(\frac{-Q}{RT}\right) \tag{8}$$

$$\dot{\varepsilon} = A[\sinh(f\!f\!\alpha)]^{n}\exp\left(\frac{-Q}{RT}\right) \tag{9}$$

The constants $n'$ and $\beta$ are determined by taking logarithms on both sides of Equations (7) and (8) [46]. These are presented as Equations (10) and (11) [22]. The plots of $\ln \dot{\varepsilon}$ versus $\ln \sigma$ (Equation (10)) and $\ln \dot{\varepsilon}$ versus $\sigma$ (Equation (11)) are used to calculate the material constants $n'$ and $\beta$ shown in Figure 7a,b [22].

$$\ln \dot{\varepsilon} = \ln A + n'\ln \sigma - \frac{Q}{RT} \tag{10}$$

$$\ln \dot{\varepsilon} = \ln A + \beta\sigma - \frac{Q}{RT} \tag{11}$$

The stress multiplier $\alpha$ is an adjustable constant. This $\alpha$-value is obtained from: $\alpha = \beta/n$ [50]. The stress exponent, n, and activation energy, Q, are determined by taking logarithms on both sides of Equation (9), which gives Equations (12) and (13) [50]. These equations are plotted as $\ln \dot{\varepsilon}$ versus $\ln (\sinh (\sigma\alpha))$ and $\ln (\sinh (\sigma\alpha))$ versus $(1000)/T$, as shown in Figure 8a,b. The average value of the slopes obtained in these plots provides n and S values for calculating the activation energy Q [25].

$$\frac{1}{n} = \left[\frac{\partial\ln[sinh(\sigma\sigma)]}{\partial(ln\dot{\varepsilon})}\right]_{T} \tag{12}$$

$$Q = RnS = Rn \left[ \frac{\partial \ln[sinh(\sigma\sigma)]}{\partial \left(\frac{1}{T}\right)} \right]_{\varepsilon} \tag{13}$$

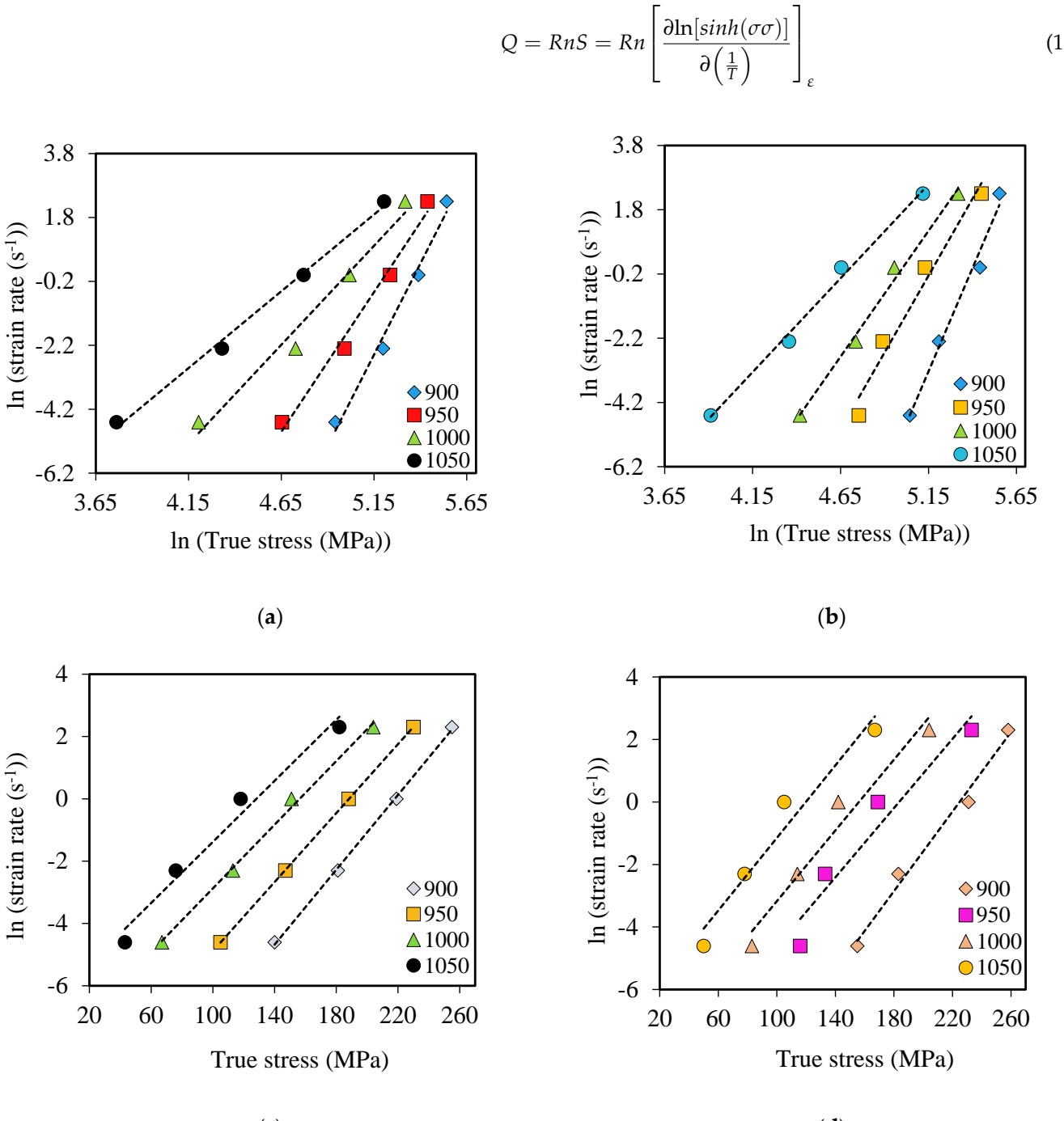

**Figure 7.** (**a**,**b**) Plots of ln ($\dot{\varepsilon}$) versus ln σ and ln ($\dot{\varepsilon}$) versus σ for the determination of $n'$ in steel A and steel B and (**c**,**d**) for determination of β in steel A and steel B.

The relationship between the flow stress and the Zener–Holloman parameter is given in Equation (6). The structure factor (A) is determined by taking logarithms on both sides of Equation (6), as in Equation (14) [21]. The *A* value is the intercept of the best fit line obtained by plotting ln (Z) versus ln (sinh(ασ)), as shown in Figure 9 [52].

$$\ln Z = \ln A + n \ln sinh(\alpha\sigma) \tag{14}$$

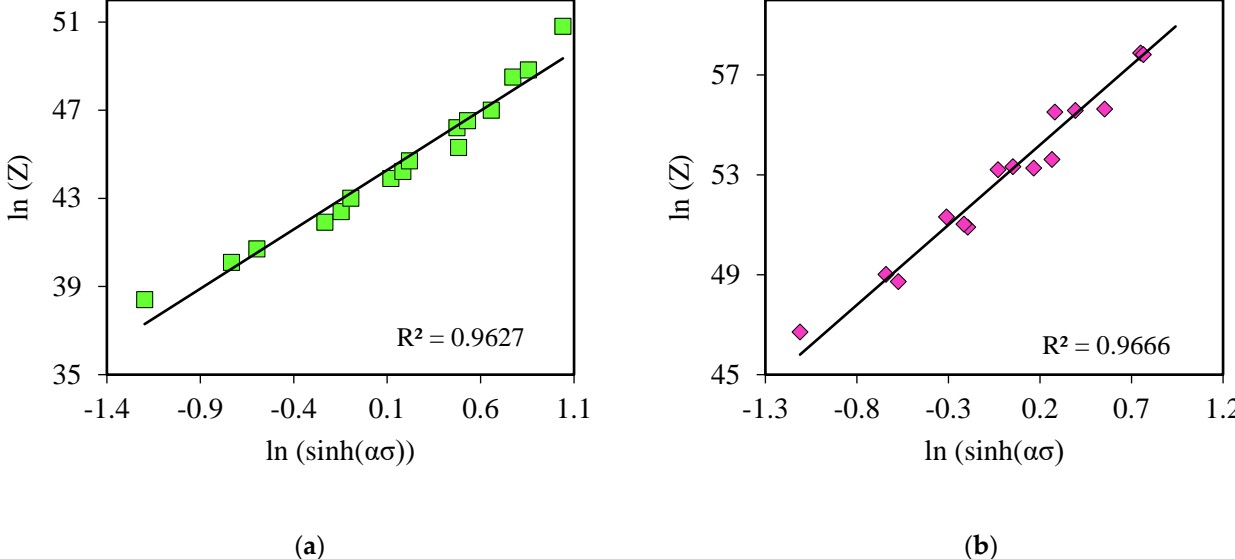

**Figure 8.** (**a**,**b**) Plots of ln $\dot{\varepsilon}$ versus σ and ln (sinh (σα)) for determination of *n* and (**c**,**d**) are plots of ln (sinh (σα)) versus (1000)/T to determine *S* value for steel A and steel B, respectively.

**Figure 9.** Plots of ln (Z) vs. ln (sinh(ασ)) for (**a**) steel A and (**b**) for steel B.

The calculated material constants for steel A and steel B for the constitutive equations are given in Table 2.

**Table 2.** Material constants and activation energy for steel A and steel B.

| P91 | $\acute{\eta}$ | $\beta$ | $\eta$ | $a$ | $S$ | $Q$ (kJmol$^{-1}$) | $ln\ A$ |
|---|---|---|---|---|---|---|---|
| Steel A | 7.79 | 0.054 | 5.76 | 0.0069 | 9.89 | 473.08 | $9.98 \times 10^{18}$ |
| Steel B | 9.03 | 0.058 | 6.67 | 0.0065 | 10.18 | 564.48 | $9.58 \times 10^{22}$ |

*3.3. Activation Energy and Stress Exponent*

Table 2 summarises the material constants $n'$, $\beta$, $n$, and $S$, and the activation energy, $Q$, for the two steels. The stress exponent and activation energy are termed as apparent values. The term apparent implies that the constitutive equation does not consider the internal microstructure changes that occur during deformation, thus assuming constant metal flow behaviour [53,54]. The stress exponent, $n$, and apparent activation energy, $Q$, indicate the deformation mechanisms that control the deformation process [45]. The activation energy shows the plastic workability of the material during the deformation process [55]. The difference between the apparent and self-diffusion activation energy of Fe in austenite provides information on the dominant softening mechanisms during forging [55]. McQueen and Ryan [56] reported that the apparent activation energy is 20% higher than the self-diffusion activation energy when DRX is the dominant mechanism. The presence of precipitates, solutes, inclusions, or reinforcements causes the activation energy to be 50% greater than the activation energy for self-diffusion. For high-stacking fault energy materials, the apparent activation energy is equivalent to or close to the activation energy for self-diffusion [41]. The results of this study show that the apparent activation energy for the two steels was 473.08 kJ/mol (steel A) and 564.48 kJ/mol (steel B). In comparison, $Q$-value for the steel A was 65% greater than the self-diffusion activation energy of Fe in austenite (284 kJ/mol), while rejuvenated creep-exhaust steel was 98% higher. These results show that the two steels exhibited low plastic deformability.

The $Q$-values of the two steels show that the rejuvenated creep-exhausted steel has a higher resistance to deformation. The reason for this might be an incomplete dissolution of carbides during deformation. The presence of carbides impedes the deformation process by hindering dislocation movement and increases the stress exponent, $n$ [52,57]. The stress exponent obtained for steel B was 6.67 and for steel A was 5.76. The stress exponent is significantly affected by an increase in flow stress. Steel B had higher flow stresses than steel A, and therefore had higher activation energy.

The stress exponent also shows the dynamic softening mechanisms Involved during forming [45]. Zhou et al. [58] reported that when dislocation climb controls deformation, the stress exponent is 5, compared to 3 for dislocation glide mechanism-controlled deformation. Dislocation climb and glide are dynamic softening mechanisms, showing that DRV occurred during deformation [59]. In this study, the flow stress–strain curves did not exhibit a clear flow stress peak. However, the flow curves exhibited a steady-state condition at higher strain, as shown in Figures 5 and 6. Lower $n$-values (<5) show that DRX was the dominant softening mechanism [48]. The stress exponent values shown in Table 1 indicate that both WH and DRV controlled the deformation process. This deformation behaviour occurs in materials with a high stacking fault energy, such as creep-resistant steel [60]. The difference between the steels may be due to the deformation conditions and steel chemistry. Higher $n$-values may be due to precipitates in the matrix, improving solid solution strengthening by pinning dislocation movement [2].

### 3.4. Constitutive Model of Flow Stresses

From materials constants and activation energy, Equations (15) and (16) [61] are the constitutive models used to predict the flow stress behaviour of steel A and steel B, respectively.

$$\dot{\varepsilon} = 9.98 \times 10^{18} [sinh(0.0069\sigma_{ss})]^{5.755175} \exp\left[\frac{-473.08}{RT}\right] \tag{15}$$

$$\dot{\varepsilon} = 9.58 \times 10^{22} [sinh(0.0065\sigma_{ss})]^{6.66945} \exp\left[\frac{-564.48}{RT}\right] \tag{16}$$

Equation (17) shows the relationship between the flow stress and Zener–Hollomon parameter Z for steel A and steel B. The flow stress for the two steels can be obtained using Equation (18) for steel A and (19) for steel B [25].

$$\sigma = \frac{1}{\alpha} ln\left\{ \left(\frac{Z}{A}\right)^{1/n} + \left[\left(\frac{Z}{A}\right)^{2/n} + 1\right]^{1/2} \right\} \tag{17}$$

$$\sigma_{ss} = \frac{1}{0.0069} ln\left\{ \left(\frac{Z}{9.98 \times 10^{18}}\right)^{1/5.76} + \left[\left(\frac{Z}{9.98 \times 10^{18}}\right)^{2/5.76} + 1\right]^{1/2} \right\} \tag{18}$$

$$\sigma_{ss} = \frac{1}{0.0065} ln\left\{ \left(\frac{Z}{9.58 \times 10^{22}}\right)^{1/6.66945} + \left[\left(\frac{Z}{9.58 \times 10^{22}}\right)^{2/6.67} + 1\right]^{1/2} \right\} \tag{19}$$

### 3.5. Verification of Constitutive Models

The accuracy of constitutive Equations (18) and (19) to predict flow stress was evaluated by comparing predicted to experimental data. Figure 10a,b show graphical plots for the two equations obtained at different strain rates and temperatures. The graphs indicate a close correlation between the flow stresses (predicted and experimental) of these two steels investigated.

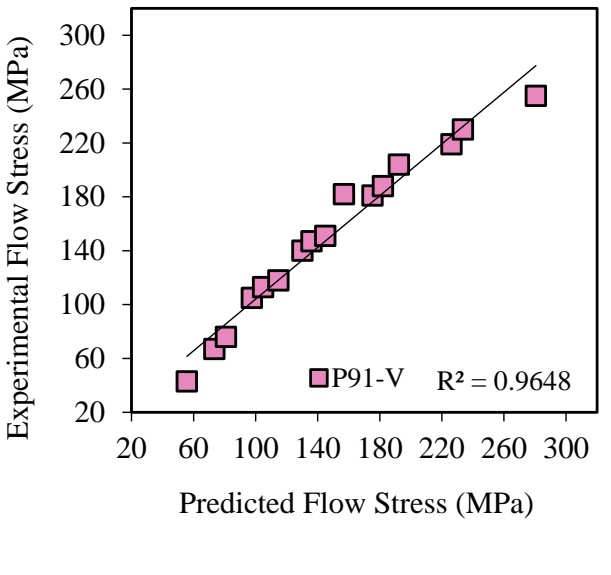

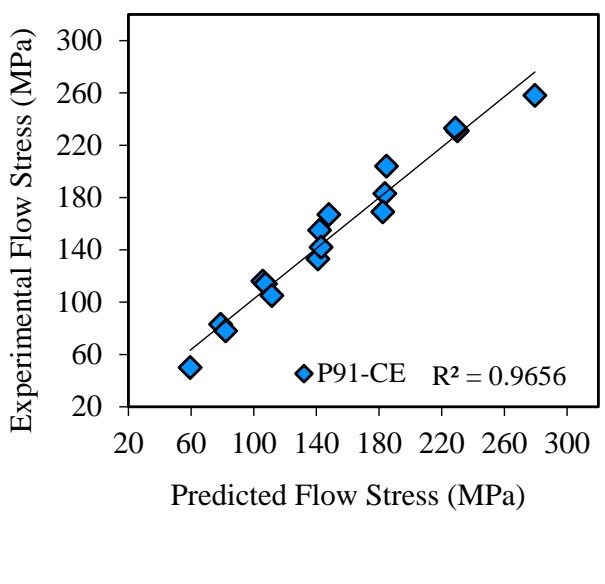

(**a**)

(**b**)

**Figure 10.** Prediction of flow stress of mathematical models for (**a**) steel A and (**b**) steel B.

The accuracy of the developed equations to predict flow stress was analysed using statistical parameters such as Pearson's correlation coefficient, *R*, in Equation (20) and average absolute relative error (AARE) in Equation (21) [62]. Pearson's correlation coefficient, *R*, ranges between 0 and 1 and is determined using Equation (20) [63]. The proximity of *R* to 1 signifies a higher capability for the model to predict flow stress and vice versa. The *R*-value obtained for steel A was 0.97, and that for steel B was 0.98. The *R*-value obtained for both materials indicated a good correlation and a higher ability of the models to predict flow stress. These values compare well with the reports of other authors, such as Mwema et al. [61] and Samantaray et al. [21], with an *R*-value of 0.994.

Pearson's correlation coefficient, *R*, is liable to bias towards greater or lower values, hence resulting in misrepresenting the accuracy of the model to predict flow stresses [61]. Using the average absolute relative error (AARE) complements Pearson's correlation method and is less subject to bias [62]. AARE analyses errors for each predicted flow stress against the experimental value on a case-by-case basis. Therefore, AARE is applied for each deformation condition to ensure the statistical efficiency of the models developed. Therefore, AARE offers better reliability and presents unbiased statistical analysis [36]. Equation (21) provides means to calculate AARE as a percentage. A lower percentage value indicates a greater ability to predict flow stresses and vice versa. The AARE obtained for the steel A was 7.62%, and for heat-treated creep-exhausted steel it was 6.54%. Reports by other authors indicated marginally lower AARE findings for the hot deformation of P91. Mwema et al. [61] obtained an AARE of 2.999%, and Samantaray et al. [21] had 5.27%. These variations are due to the differences in hot deformation parameters in the respective studies. The study by Mwema et al. [61] used a hot deformation temperature between 900 °C and 1200 °C and strain rates of 1 s$^{-1}$, 5 s$^{-1}$, 10 s$^{-1}$ and 15 s$^{-1}$. In Samantaray et al. [21], the temperature for hot deformation was between 850 °C and 1100 °C and the strain rates were 0.001 s$^{-1}$, 0.1 s$^{-1}$ and 100 s$^{-1}$. Validation by both models thus signified satisfactory levels of confidence in the respective constitutive equations to predict flow stresses.

$$R = \frac{\sum_{i=1}^{N}(P_i - P)(E_i - E)}{\sqrt{\sum_{i=1}^{N}(P_i - P)^2}\sqrt{\sum_{i=1}^{N}(E_i - E)^2}} \tag{20}$$

$$AARE = \frac{1}{N}\sum_{i=1}^{N}\left|\frac{E_i - P_i}{E_i}\right| \times 100\% \tag{21}$$

where $E_i$ is the flow stress from the experiment, $P_i$ is the flow stress predicted from constitutive equations, and $E$ and $P$ are the average flow stress values of experimental data and the predicted data, respectively. $N$ is the sum of data points.

### 3.6. Comparison of Constitutive Equations

The mathematical model developed for steel A was used to predict flow stresses of steel B and vice versa. The analysis intended to determine the suitability of having a model that can be applied to predict the flow stresses of either of the two steels investigated. Figure 11a is a plot of experimental stress versus the calculated flow stress where a model of steel A is used to predict the flow stress of steel B. From the statistical analysis, Pearson's correlation coefficient, *R*, was 0.9637, and AARE was 7.19%. Hence, there was a good correlation between measured and predicted flow stress. The model for heat-treated creep-exhausted P91 steel was also used to predict the flow stress for steel A (Figure 11b). The plot shows a good correlation between the measured and predicted flow stresses. Pearson's correlation coefficient was 0.95, and AARE was 8.36%. From the analysis, the steel A model had a better approximation than the steel B model in predicting the flow stress for either P91 steel.

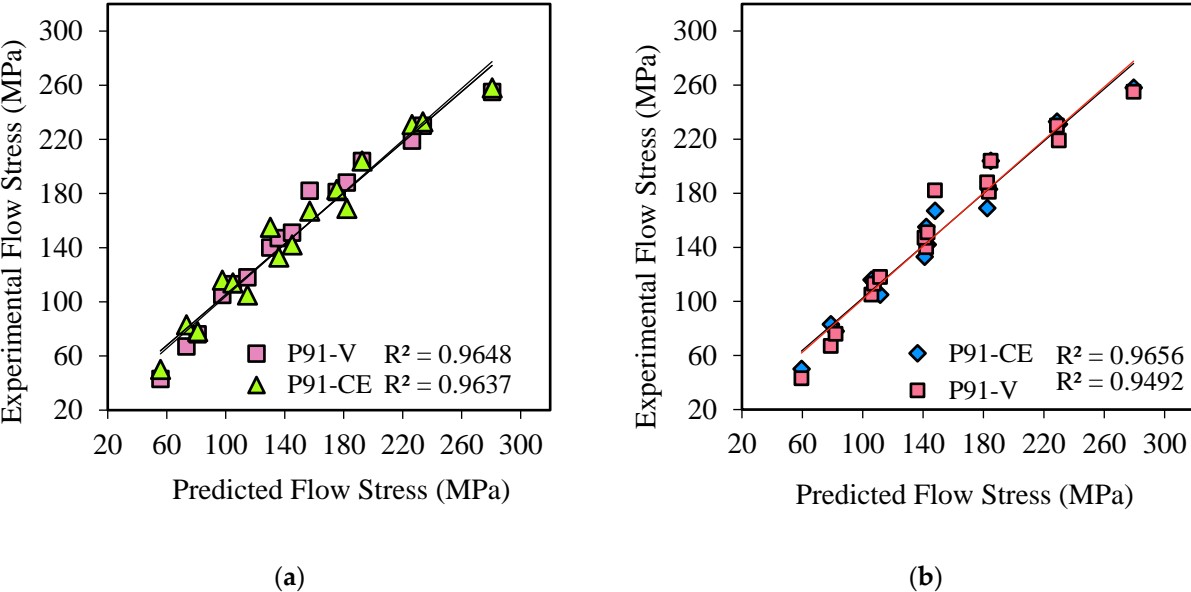

(**a**)  (**b**)

**Figure 11.** (**a**) Steel A model used to predict flow stresses of steel B. (**b**) Steel B model to predict flow stresses in Steel A.

## 4. Conclusions

In this study, virgin (Steel A) and creep exhaust steel (Steel B) were subjected to various deformation conditions (temperature 900–1050 °C and strain rates of 0.01–10 s$^{-1}$), and the following conclusions were drawn:

1. The flow stress–strain curves show that the flow stress increased with an increase in strain rate (0.01 s$^{-1}$ to 10 s$^{-1}$) and decreased with an increase in temperature (900 °C to 1050 °C) for the two steels. The flow stress–strain curves exhibited a DRV+WH as the deformation mechanism.
2. The apparent activation energy of steel A was 473.08 kJ/mol, and for steel B it was 564.48 kJ/mol. These Q-values were much higher compared to the self-diffusion energy of iron in austenite (270 kJmol$^{-1}$)
3. The mathematical constitutive models for steel A and steel B to a total strain of 0.6 are as given in Equations (22) and (23).

$$\dot{\varepsilon} = 9.97885 \times 10^{18}[sinh(0.006916\sigma_{ss})]^{5.755175}\left[\frac{-473.0843}{RT}\right] \tag{22}$$

$$\dot{\varepsilon} = 9.58432 \times 10^{22}[sinh(0.00647\sigma_{ss})]^{6.66945}\left[\frac{-564.4791}{RT}\right] \tag{23}$$

4. The constitutive models were validated using Pearson's correlation coefficient, *R* and the average absolute relative error, AARE. For steel A, *R* was 0.97, and AARE was 7.62%. For Steel B, *R* was 0.98, and AARE was 6.54%. The developed models were used interchangeably with acceptable accuracy. Using the model for steel A on steel B, *R* was 0.96, and AARE was 7.19%. Similarly, using the steel B model on steel A, the *R* was 0.95, and AARE was 8.36%.

**Author Contributions:** Conceptualization, S.M., J.O., J.V.d.M., F.M., M.B. and D.K.; methodology, M.B., J.O., F.M. and M.B.; investigation, S.M., J.O., J.V.d.M., M.B. and D.K.; writing—original draft preparation, S.M.; writing—review and editing, S.M., J.O., M.B. and D.K. All authors have read and agreed to the published version of the manuscript.

**Funding:** This research received no external funding.

**Institutional Review Board Statement:** Not applicable.

**Informed Consent Statement:** Not applicable.

**Data Availability Statement:** The raw data presented in this study are available on request.

**Conflicts of Interest:** The authors declare no conflict of interest.

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
