# Peer review of "Comparative Study on Hot Metal Flow Behaviour of Virgin and Rejuvenated Heat Treatment Creep Exhausted P91 Steel"

_applsci, doi:10.3390/app13074449_

Round 1

Reviewer 1 Report

Review report Comparative study on hot metal flow behaviour of virgin and rejuvenated heat treatment creep exhausted P91 steel. Work is quite interesting and presented in well manner. The comments are listed below:

1.       Abstract section is prepared well.

2.       Add some discussion related to application point of view.

3.       Introduction part is prepared well but can be strengthen by adding some recently published work on CSEF P91/P92 steel: https://doi.org/10.1007/s43452-022-00592-5; https://doi.org/10.1016/j.ijpvp.2022.104782.

4.       In material and methods section following correction is required:

a.        Discuss about the selection of the heat treatment parameters or add proper references.

b.       Add mechanical properties and composition of the material.

c.        Add SEM image with EDS spectra.

5.       Results and discussion:

a.        How was the true strain calculated?

b.       Add the references for each equation.

c.        Technical discussion related to variation in strain and strain is missing.

d.       The discussion of flow stress in terms of the microstructure is also an essential part. Add some discussion related to it.

6.       Shorten the conclusion section. Keep only key points. 

Author Response

The author's responses are attached.

Reviewer 2 Report

Comments and specific questions requiring clarification:

1.       I recommend not to divide the words. If it is possible please correct it. It applies to the whole manuscript.

2.       Show please chemical composition of used steel at table form.

3.       Please correct repeated errors in references to the literature.

4.       Why did You made compression test with 0.6 strain value only? It is possible to obtain a little more strain value (1.2) with this machine. Explain it for me please and if it is possible add this information to the paper also please.

5.       Reffers to the text between line no. 139-147: Did You use special grease and pads between tool and material surface? If not – explain it why and add this information to the paper. It is cecessary in my opinion. Special grease and pads between tool and material surface helps to make strain more uniform in the material. Moreover it helps also decrease friction effect (barrel). Then any correction isn’t necessary.

6.       Reffers to the figure no. 4: add please information about strain rate to the strain-stress curves presented at fig. no.4. It is necessary in my opinion.

7.       There are also others constitutive equations which describes material flow as a function of strain, strain rate and temperature – for example Hensel-Spittel equation and others which are commonly used in FEM software with good accuracy. Please add information (one paragraph) about these models to Your paper. Moreover add the information why did You decided to use exactly Arrhenius model?

Author Response

The author's responses are attached.

Round 2

Reviewer 2 Report

Thank You for the answer, but there are still errors in literature references. Please correct repeated errors in references to the literature.

Author Response

Dear Reviewer

Please attached are our responses to your comments.
